# Selective Blocking of Graphene Defects Using Polyvinyl Alcohol through Hydrophilicity Difference

**DOI:** 10.3390/ma16052001

**Published:** 2023-02-28

**Authors:** Yoon-jeong Kim, Yang Hui Kim, Seokhoon Ahn

**Affiliations:** 1Institute of Advanced Composite Materials, Korea Institute of Science and Technology (KIST), Jeonbuk 55324, Republic of Korea; 2School of Semiconductor and Chemical Engineering, Jeonbuk National University, Jeonbuk 54896, Republic of Korea

**Keywords:** graphene, defect healing, polyvinyl alcohol (PVA), hydrophilic interactions

## Abstract

Defects on graphene over a micrometer in size were selectively blocked using polyvinyl alcohol through the formation of hydrogen bonding with defects. Because this hydrophilic PVA does not prefer to be located on the hydrophobic graphene surface, PVA selectively filled hydrophilic defects on graphene after the process of deposition through the solution. The mechanism of the selective deposition via hydrophilic–hydrophilic interactions was also supported by scanning tunneling microscopy and atomic force microscopy analysis of selective deposition of hydrophobic alkanes on hydrophobic graphene surface and observation of PVA initial growth at defect edges.

## 1. Introduction

Polymer–based graphene nanocomposites hold great potential for applications such as sensors, energy harvesting, and gas barrier films because of their superior mechanical, optical, and barrier properties [1,2,3,4,5,6]. However, defects on graphene limit further development and commercialization [7,8,9,10,11,12,13,14]. The graphene defects could be generated from synthetic or artificial processes. For example, graphene synthesized through the chemical vapor deposition (CVD) method generally has countless defects with various sizes ranging from nanometer to micrometer [15,16,17,18,19,20]. There could be defects that have not been synthesized, or perfect hexagons could not be formed at grain boundaries where graphene domains with different crystal directions meet. Thus, the honeycomb structure may not be formed, and defects with various shapes and sizes such as pentagon and octagon may occur. In addition, the mismatch of grain boundaries could also lead to overlapped defects at grain boundaries [8]. Since graphene layers are physically connected, defects could be a pathway of gas molecules, limiting gas barrier application using CVD graphene.

Furthermore, graphene defects do not occur only in the synthesis process. Defects may also occur in the process of transferring graphene to the target substrate [21]. Although graphene is considered a very high–strength material at the atomic level, graphene is very weak at the macroscopic level. Therefore, defects are likely to occur due to inexperienced handling in the process of transfer. If graphene is transferred onto another substrate while the metal is not completely melted during the etching process, a metal mass may be placed under the graphene and tension may be applied to the graphene, which might cause defects. Graphene is a theoretically perfect material as a gas barrier; however, as described so far, perfect graphene does not exist in reality. Therefore, defects are a very critical issue in developing graphene–based applications [22,23,24].

A number of graphene healing methods have been reported. For example, since DFT calculations or molecular dynamics could study the interactions of graphene with a variety of molecules, these methods hold great potential to design graphene–defect–healing processes [25,26,27]. As another example, defects on graphene–like surface graphite at 625 K can be healed with flowing hydrocarbons or acetylene. As hydrocarbons decompose at 100–150 °C [28], carbon enters and fills the empty space of the defect, bonds with other nearby carbon, and the defect is healed [29]. Similarly, when hydrogen or oxygen flows, gas molecules are adsorbed to carbon in the part connected to the pentagons in the 7–5–5–7 defect. After that, the defect is healed as it exits with two carbon atoms [30]. Another way of providing carbon sources into graphene defects is by using another graphene layer. When there are multiple layers of graphene, adjacent layers are used as a carbon source. Carbons in one layer could move to defects in the other layer at 2500 K and defects in the other graphene layer could be healed [31]. An adsorption healing method using carbon monoxide and nitrogen monoxide was also reported. It involves sequentially exposing carbon monoxide and nitrogen monoxide to graphene [32]. After the first flowing carbon monoxide, carbon monoxide is adsorbed in the graphene defect. Then, the oxygen of carbon monoxide is reacted with the second flowing nitrogen monoxide to become nitrogen dioxide. Therefore, graphene defects could be filled with carbon from carbon monoxide. Catalytic substrates could be used to heal graphene defects. Graphene synthesized on nickel could contain thermodynamically unstable ring structures such as pentagons and octagons. These unstable defects could be rearranged to a hexagon via heat treatment on catalyst substrates [33]. Electrochemical deposition was used to selectively heal defects by using metal atoms because nucleation of metal atoms during electrochemical deposition occurs at graphene defects [34]. However, these healing processes may not be suitable for industrial applications upon considering high process cost and mass production. For example, the high temperature treatment process is difficult to apply in the industry because it greatly increases the process cost. In addition, the method of healing graphene defects with metals has disadvantages of losing intrinsic properties of graphene such as flexibility and transparency, and not covering micrometer size defects. Therefore, developing a healing method suitable for mass–production processes with a low cost of the process is still required from an industrial–application standpoint. Although it was demonstrated that self–assembled alkane molecules were able to physically block graphene defects smaller than alkane size, over micrometer size defects could not be selectively healed [24].

Here, we report that graphene defects over micrometer could be selectively filled with hydrophilic polymers using hydrophilic–hydrophilic interactions. Although the graphene surface is hydrophobic, graphene defects are generally hydrophilic due to the existence of hydrophilic functional groups such as hydroxyl and carboxyl groups [35,36,37]. Upon considering these differences, we hypothesized that graphene defects could be selectively filled with hydrophilic polymers which are able to make hydrogen bonding with hydrophilic functional groups at graphene defects. In order to demonstrate this hypothesis, artificial graphene defects measuring 5 μm were periodically created using photolithography. On this patterned graphene, alkane as a hydrophobic material and polyvinyl alcohol (PVA) as a hydrophilic material were deposited in order to prove the hypothesis. As a result, alkane materials were only deposited on graphene surface rather than defects, whereas PVA was deposited in graphene defects by forming hydrogen bonding between hydroxyl groups in PVA and hydrophilic functions such as hydroxyl or carboxylic groups from graphene defects as described in Figure 1.

## 2. Materials and Methods

### 2.1. Materials and Instruments

Polyvinyl alcohol (PVA, Mw 146,000–186,000, Tg 74.3 °C, Tm 222.7 °C (Appendix A), Sigma–Aldrich, St. Louis, MO, USA), hexatriacontane (HTC (C_36_H_74_), Sigma–Aldrich, St. Louis, MO, USA), and heptane (C₇H₁₆, Sigma–Aldrich, St. Louis, MO, USA) were used without a further purification process. As a solvent, ethanol was purchased from Sigma–Aldrich. The height of PVA in defects on graphene was obtained using AFM (Park system, NX10) in non–contact mode.

### 2.2. Synthesis of Graphene

Graphene was synthesized by following the chemical vapor deposition (CVD) method [18,23,38]. This method is generally well–known and widely used to obtain large–area graphene for various applications. The cut Cu foil was placed in the furnace and the vacuum was held. While flowing hydrogen gas for 1 h 30 min, the temperature increased to 1030 °C. Then, 5 sccm of methane gas flowed for 1 min to minimize multiple nucleation sites and then 13 sccm of methane gas flowed for 8 min to grow graphene. After that, the hydrogen flow was lowered to 15 sccm from 100 sccm, and quickly cooled to room temperature.

### 2.3. Graphene Transfer on Si Wafer

Graphene was transferred onto the Si wafer through the wet transfer method using polymethylmethacrylate (PMMA) as a supporting material [22,39]. The wet transfer method is also one of the widely known methods of transferring large–area graphene to Si wafer for application. First, graphene synthesized with CVD was fixed on a polyethylene terephthalate (PET) film with 3M tape, and PMMA was spin–coated on the graphene. The coated PMMA/graphene was soft baked in a hot plate or oven (for 1 min at 180 °C for hot plate/for 30 min at 60 to 70 °C for oven). Graphene was etched with O_2_ plasma using reactive ion etching (RIE) equipment in order to remove graphene off the back side by inverting the graphene coated with PMMA. To obtain only graphene from CVD–graphene, Cu foil was etched in an aqueous ammonium persulfate (APS) solution for 4 h. Graphene floating on the APS solution was transferred to deionized (DI) water using PET film and the remaining APS solution was rinsed three times. Before graphene was transferred to Si wafer, the Si wafer was cleaned in acetone and isopropyl alcohol (IPA) using sonication for 10 min, respectively. Then, graphene was transferred to the target substrate and dried at room temperature for 30 min in order to prevent water from being trapped between the graphene and the Si wafer substrate. After that, it was put oven for an hour, dried completely, and the PMMA used as a supporting material in acetone for an hour on hot plate at 40 °C was removed. Graphene on Si wafer was rinsed in IPA and extra solvent was blown using nitrogen.

### 2.4. Fabrication of Graphene Defects

Holes were manufactured using photolithography [40,41]. A photoresist (PR) solution was spin–coated on the transferred graphene and baked at 90 °C for 2 min. The sample was exposed to UV light for 7 s using a mask with holes with a size of about 5 μm and the chain was broken in PR at that time. The sample was put in developer solution for 40 s to produce an artificial defect. When the PR disappeared in the developer solution, the site of graphene exposed to UV light disappeared with PR. Then, the sample was rinsed in DI water. Extra PR was removed in acetone. This process was illustrated in Figure 2.

### 2.5. Preparation of HTC and PVA Solutions

HTC heptane solution of 1 mM was prepared by ultra–sonicating the HTC and heptane mixture for 1 h at room temperature. For the PVA solution, PVA was dissolved in ethanol with a concentration of 0.1 g/mL by stirring the solution for 30 min at 45 °C.

### 2.6. Deposition of HTC and PVA

For HTC deposition, the HTC solution in heptane (b.p 98 °C) was dropped on patterned graphene with holes, and the solvent was slowly vaporized at room temperature to obtain multilayer HTC on graphene. For the PVA deposition, the PVA solution in ethanol was dropped on prepared graphene with a hole. Then, the ethanol solvent was slowly vaporized at room temperature and quenched before fully vaporized by blowing the solution to investigate the initial growth of PVA. For the full deposition of PVA, after dropping PVA solution on patterned graphene, the ethanol solvent was slowly and completely vaporized at room temperature. Then, the sample was annealed on a hot plate at 210 °C for 10 min to prepare the selectively deposited PVA films in graphene defects where Tg and Tm of PVA are 74.3 °C and 222.7 °C, respectively.

### 2.7. AFM Measurement

Samples for AFM measurements were created by making artificial defects on CVD–graphene transferred on Si wafer substrate and dropping the HTC or PVA solutions as shown in the experimental methods 2.2, 2.3, 2.4, 2.5, and 2.6 above. The AFM imaging process was conducted in non–contact mode using park system’s NX10 equipment. Images were scanned measuring 10 μm × 10 μm or 20 μm × 20 μm in size. The values of the scan rate for the pristine graphene sample, graphene coated with multilayer HTC, and PVA in defects on graphene were 0.35 Hz, 0.6 Hz, and 0.3 Hz, respectively. The values of Z servo gain for the pristine graphene sample, graphene coated with multilayer HTC, and PVA were 1, 1.5, and 1, respectively.

### 2.8. Scanning Tunneling Microscopy (STM) Study of Self–Assembled HTC on Graphene

A self–assembled HTC monolayer on CVD–graphene was imaged using STM (Bruker Singapore Pte. Ltd., Singapore). The STM tip was created by cutting the tip of a wire made of Pt/Ir (California Fine Wire co, Grover Beach, CA, USA, 80%/20%) alloy. In order to investigate a self–assembled HTC monolayer at molecular level, HTC was dissolved in 1–phenyloctane for STM study. The HTC solution was replaced with CVD–graphene on copper substrate. The STM tip went into the solution and approached the graphene surface. The self–assembled structure of HTC was mainly scanned at 700~800 mV and 200~300 pA. All image processing was performed using SPIP software.

## 3. Results and Discussion

### 3.1. HTC Deposition on Patterned Graphene

Alkane molecules, hexatriacontane (HTC), were deposited on patterned graphene in order to confirm that hydrophobic molecules prefer to be deposited on only a hydrophobic graphene surface rather than hydrophilic graphene defects as shown in Figure 3. HTC would make self–assembled layers on graphene whenever HTC solution is replaced on graphene surface due to enough stabilization energy from van der Waals interactions from HTC–HTC and HTC–graphene. In the solution, HTC molecules prefer to randomly exist due to the thermodynamic rule of entropy increase. However, when HTC makes a van der Waals contact with the graphene surface, HTC would be nucleated on graphene due to strong van der Waals interaction and start to make a self–assembled monolayer [24]. Because the enthalpy reduction from HTC molecules in a self–assembled monolayer and HTC–graphene contact overcomes the barrier of entropy, Gibbs free energy turns to negative, allowing the formation of an HTC monolayer on graphene surface even in the solution, as shown in Figure 4. The synthesized CVD graphene surface and zoomed–in graphene honeycomb structure (inset) is shown in Figure 4a. Under the HTC solution, a self–assembled HTC monolayer is visualized using STM. The steps originate from the crystalline copper surface and the lines are from the columnar structure of alkanes (Figure 4b). The zoomed–in image shows that alkane molecules are stacked by forming the columnar structure through van der Waals interactions (Figure 4c). The slow evaporation of the solvent would lead to three–dimensionally stacked HTC films on graphene. Interestingly, this phenomenon would not occur on amorphous SiO_2_ surface. The lack of a strong van der Waals interaction between HTC and surface leads to HTC molecules not being nucleated on the amorphous SiO_2_ surface in the early nucleation stage. Therefore, it was expected that HTC would only be deposited on patterned graphene and not on the amorphous SiO_2_ surface. This hypothesis is experimentally demonstrated using patterned graphene. The graphene has periodically aligned 5 μm defects as shown in Figure 3a. On this graphene, HTC was deposited through slow evaporation of heptane solution (Figure 3b). After the deposition, the thickness and morphology of HTC films on graphene were investigated using AFM as shown in Figure 3c. The initial height from SiO_2_ to graphene before the deposition was measured as ~2 nm through the non–contact mode of AFM (Figure 2b). After the deposition, the thickness of the HTC film was 60 nm (Figure 3c blue line) and the morphology became much smoother than that of the graphene surface (Figure 3c red line). Interestingly, as shown in Figure 3b, HTC was only deposited on the graphene surface rather than in defects. This result supports the hypothesis that hydrophobic molecules prefer to be deposited on a hydrophobic graphene surface, presumably due to hydrophobic–hydrophobic interactions. In other words, hydrophilic defects might be filled with hydrophilic molecules through hydrogen–bonding interactions.

### 3.2. PVA Deposition on Patterned Graphene

In order to prove the hypothesis that hydrophilic–hydrophilic interaction between PVA and defects allows selective deposition, the initial growth of PVA was investigated before completing deposition. As shown in Figure 5, PVA particles were only located at the defect edges at initial nucleation stages. Interestingly, no PVA particles were observed on the hydrophobic graphene surface, supporting the hypothesis that PVA prefers to interact with hydrophilic functional groups at defect edges such as carboxylic acid and hydroxyl group, as described in Figure 5c [35]. This result indicates that hydroxyl groups in PVA make hydrogen bonding with defective edges as described in Figure 5c. After completing the deposition of PVA, it was observed that PVA was periodically deposited only in graphene defects and not on graphene surface, as shown in Figure 6a. The enlarged image (Figure 6b) clearly shows that only defects are filled with PVA, although the height and shape of PVA are slightly different. The thickness and morphology of the deposited PVA were investigated using AFM (Figure 6c). The thickness was 250 nm and the mountain–like morphology was observed. The formation of this mountain–like morphology also indicates that PVA does not prefer to be deposited on the hydrophobic graphene surface. Therefore, the hydrophilic nature of defect edges and PVA, and the hydrophobic nature of the graphene surface allow PVA to be selectively deposited in graphene defects. This mechanism study of selective deposition of polymers in graphene defects would provide insights to the development of applications such as graphene–based gas barriers and various polymer nanocomposites.

## 4. Conclusions

Although various healing methods for graphene defects have been developed, defects over micrometer size are not efficiently healed. Upon considering the hydrophobic surface of graphene and the hydrophilic characteristics of graphene defects, it was demonstrated that hydrophilic polymer could selectively fill graphene defects over micrometer size through hydrogen bonding with functional groups at defect edges. This mechanism study for selective deposition of polymers in graphene defects would provide insights for further development of polymer–graphene composites.

## Figures and Tables

**Figure 1 materials-16-02001-f001:**
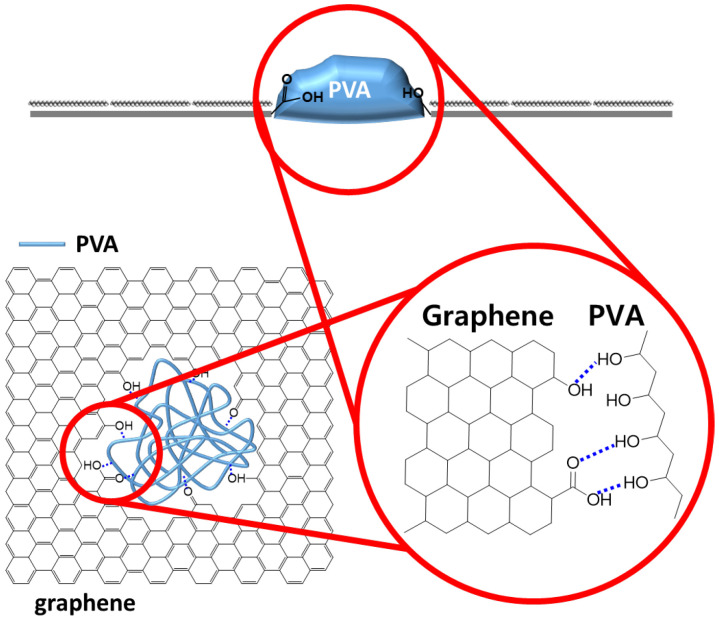
Scheme of blocking graphene defects with PVA by forming hydrogen bonding.

**Figure 2 materials-16-02001-f002:**
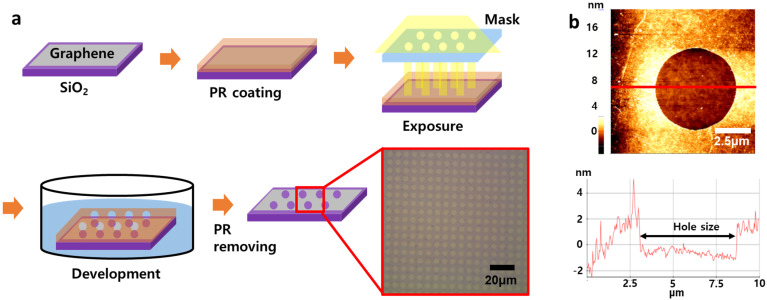
Fabrication processes for micrometer–sized defects. (**a**) Processes of making defects in graphene through photolithography, and (**b**) AFM image of defect with a line profile.

**Figure 3 materials-16-02001-f003:**
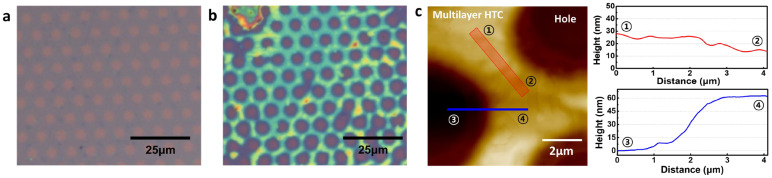
Multilayer HTC on graphene. (**a**) Pristine graphene. The holes were made by photolithography, (**b**) heat treatment at 100 °C for 20 min after coating with multilayer HTC on graphene, and (**c**) AFM image and height information of coated with multilayer HTC on graphene.

**Figure 4 materials-16-02001-f004:**
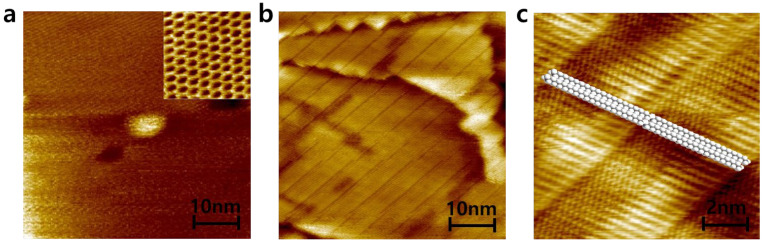
STM images of (**a**) CVD–graphene and zoomed–in graphene (inset), (**b**) a self–assembled HTC monolayer on CVD–graphene, and (**c**) zoomed–in HTC monolayer with overlay.

**Figure 5 materials-16-02001-f005:**
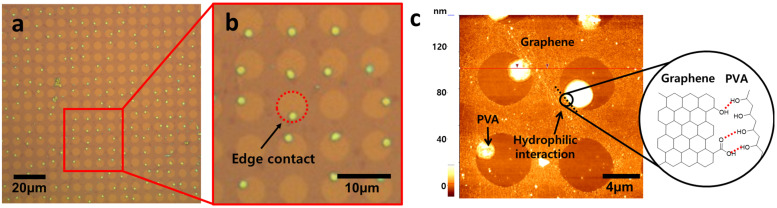
(**a**) OM image of PVA in the defects of graphene, (**b**) the OM image of the PVA particles in contact with the edge of the defects, and (**c**) AFM image of PVA in the defects of graphene.

**Figure 6 materials-16-02001-f006:**
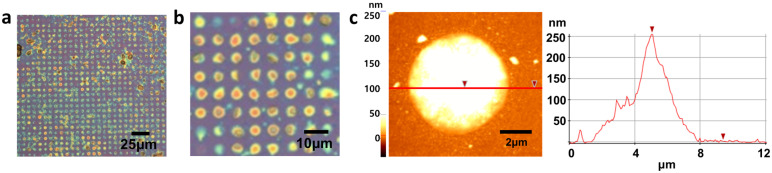
(**a**) OM image of PVA deposition on patterned graphene, (**b**) the enlarged image of (**a**), and (**c**) AFM image of PVA in the defect of graphene. The height of the hole–filled PVA was about 250 nm.

## Data Availability

Not applicable.

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
