# Peer review of "Selective Blocking of Graphene Defects Using Polyvinyl Alcohol through Hydrophilicity Difference"

_materials, 2023, doi:10.3390/ma16052001_

Round 1

Reviewer 1 Report

Dear Authors/Editor,

Thank You very much for the trust and opportunity to revise scientific paper entitled: “Selective blocking of graphene defects using polyvinyl alcohol through hydrophilicity differencewritten by Yoon-jeong Kim , Yang Hui Kim and Seokhoon Ahn for Materials journal.

In my opinion the article is very interesting and has very good scientific quality. The topic is very modern, popular moreover presented results and conclusions are very valuable for research groups working with graphene as well as for graphene production and possible application. Results, comments and conclusions are quite well documented and supported with literature. Moreover, It is written in understandable style with good English language.

If it is possible I would like to recommend to authors that they should add some information and results about chemical structure (FTIR spectroscopy for example) after PVA application and deposition on graphene to confirm proposed mechanism/reaction between graphene layer and PVA.

Moreover I have some doubts what chemical structures changes can occur after the heating (to 210°C) process of PVA on graphene layer.

In general, therefore, I recommend publishing this article after the above-mentioned addition.

Best regards

Reviewer

Reviewer 2 Report

This manuscript reports on the selective blocking of graphene defects by employing polyvinyl alcohol molecules as based on the hydrophilicity difference. Authors conclude that PVA selectively filled hydrophilic defects on graphene after the deposition by solution process. The research results also indicate that the mentioned mechanism of the selective deposition is supported by selective deposition of hydrophobic alkanes on hydrophobic graphene surface and observation of PVA initial growth at defect edges. Thus, clear results backed by AFM measurements, as well as STM are provided. These are novel results for concerning defects in graphene, and they are very well transmitted by the discussion in the present manuscript.

The research questions as defined in this work are much worth investigation while in the present context the measurement efforts (including STM) as employed are not only adequate for the present purpose, but highly sophisticated while also the interpretation of characterization results seems correctly done.

All in all, the results are interesting and presented in a way which is easy and valuable for a wider audience of readers. All these results seem like worthy points of departure for the discussion and are plausible and well substantiated in the view of the conclusions drawn.

The reported results bring new knowledge and certainly represent an original contribution with probable wide impact in the field.

The authors chose an adequate structure of the manuscript for such a study. Also, they provided a balanced realistic and nicely illustrated presentation of their results and corresponding analysis that is of much scientific and practical interest and adds new knowledge to the field.

The present manuscript is a significant contribution, this work once published would be instructive and suggestive in terms of further studies and with good chances be cited.

There are some minor issues with this already excellent manuscript that will need to be addressed before the manuscript becoming suitable for publication, i.e., it can be considered for publication after a minor revision:

1: Authors do not mention in the abstract what kind of measurements/characterization they use. It would be advisable to explicitly mention these aspects which will contribute for the clarity of the abstract.

2: Authors mention several times “high temperatures”, “low temperatures”. It would be advisable to include concrete temperature values in these cases which may be contextualized by comparison to (temperatures as relevant to) similar processing.

3: Since defects in graphene happen in a wide variety, maybe some short systematics should be at least mentioned.

4: In the introduction, the authors miss that interactions of graphene with variety of molecules can be studied (and guided) by DFT calculations as in [Carbon 81 (2015) 620-628] but even also by (ab initio) Molecular Dynamics as in [Physical Chemistry Chemical Physics 25 (2023) 829-837] with direct practical implications for the credibility of the claims about healing of defects, etc. This aspect should be acknowledged in the present manuscript.

5: Spell-check and stylistic revision of the paper are necessary. Some long sentences, as well as misspellings, etc., are noticeable throughout the text.
